# A Rapid Immunochromatographic Method Based on Gold Nanoparticles for the Determination of Imidacloprid on Fruits and Vegetables

**DOI:** 10.3390/foods12030512

**Published:** 2023-01-23

**Authors:** Steven Suryoprabowo, Aihong Wu, Liqiang Liu, Hua Kuang, Chuanlai Xu, Lingling Guo

**Affiliations:** State Key Lab of Food Science and Technology, School of Food Science and Technology, Jiangnan University, Wuxi 214122, China

**Keywords:** gold immunochromatography test, imidacloprid, monoclonal antibody, strip test

## Abstract

Imidacloprid (IMP) is toxic and a potential carcinogen that is most widely used as an insecticide for pest control and seed treatment. It is important to produce a rapid and sensitive assay for on-site monitoring. We have developed a novel lateral flow assay (LFA) using a sensitive monoclonal antibody (mAb) for monitoring IMP residues on fruits and vegetables. The 50% inhibition concentration result that was found when using the ELISA method was 0.247 ng mL^−1^, with the cut-off limits using the LFA method the result was 10 ng mL^−1^ (0.01 M PBS), and in the samples it was 20 ng mL^−1^ (with a recovery rate of 96–104.7% for Chinese cabbage, cowpea, apple, and pear samples, respectively). All of the results can be determined within seven minutes. The proposed LFA method is a valid, quick, and stable assay for the on-site detection of IMP in large numbers of samples.

## 1. Introduction

Imidacloprid (IMP, Figure 1a) is a recently developed neonicotinoid insecticide that is a highly effective to persistent organochlorine-based pesticide with a broad spectrum of activity and a low potential for bioaccumulation [1]. The mode of action of IMP involves interference with the signal transmission of the insect nervous system [2]. Due to its polarity, the extensive application of IMP, and its residues on fruits and vegetables, pose a high risk to consumers. Residual pesticides pose potential hazards to the environment and agricultural products, as the residual metabolites are toxic. Therefore, it is essential to develop a fast and sensitive method to determine analytes, which is particularly important for on-site monitoring in order to ensure food safety [3,4,5].

The various methods that are currently used to detect pesticide residues on agricultural products include high-performance liquid chromatography (HPLC) [6,7,8], gas chromatography (GC) [9,10,11], thin-layer chromatography (TLC) [12,13], and liquid chromatography–tandem mass spectrometry (LC–MS/MS) [14,15]. Chromatographic techniques are the most commonly used tools by the virtue of their wide application and high sensitivity, however, they are expensive and require skilled operators [16,17,18].

Numerous enzyme-linked immunosorbent assays (ELISAs) based on antibody–antigen interactions are quite popular for the analysis of pesticide residue in food and environmental samples. A direct extraction procedure is typically used to target pesticide-contaminated solid samples in order to shorten the procedural duration of an ELISA, as compared to conventional chromatography [19,20].

ELISA is a sensitive method for the detection of pesticides, including IMP. One study [21] reported the use of an ELISA method for the detection of IMP and thiamethoxam that employed a simple dilution method in honey, without extraction or cleanup, with limits of detection of 20 and 5 ng g^−1^, respectively. A commercially available ELISA kit for the detection of IMP residue on agricultural samples (apple) reported a LOD of 0.5 ng mL^−1^ [22]. However, the ELISA method requires a washing step and incubation for an enzyme–substrate reaction, therefore it is not proper for on-site determination [23,24].

Recently, lateral flow assays (LFAs) have been widely used for detection in food safety, environmental monitoring, disease diagnosis, and other applications, and have many advantages, as they are rapid, simple to operate, inexpensive, and sensitive [25,26,27]. Most LFAs use antibodies that are labelled with gold nanoparticles (AuNPs), while others employ fluorescent microspheres as an alternative, due to the advantages of long emission wavelengths, an extended shelf-life, and excellent photostability [28,29,30].

An LFA is a reliable and rapid detection method that has become increasingly popular. To the best of our knowledge, this is the first study to report the use of an LFA assay to detect IMP residues on fruits and vegetables. In this study, the sensitivity and the accuracy of the LFA method for the detection of IMP residues have been tested using Chinese cabbage, cowpea, pear, and apple samples.

## 2. Materials and Methods

### 2.1. Chemical Reagents and Instruments

The IMP was purchased from J&K Scientific Ltd. (Shanghai, China). The gelatin was acquired from Beijing Biodee Biotechnology Co., Ltd. (Beijing, China). The other chemical reagents, including complete and incomplete Freund’s adjuvant, enzyme immunoassay-grade horseradish peroxidase-labeled goat anti-mouse immunoglobin, N-hydroxysuccinimide (NHS), bovine serum albumin (BSA), keyhole limpet hemocyanin (KLH), 1-etyl-3-(3-dimethylaminopropyl) (EDC), and Tween-20 were bought from Sigma (St. Louis, MO, USA). The reagents for the cell experiments were purchased from Nanjing Sunshine Biotech Co., Ltd. (Nanjing, China).

The ELISA plates were acquired from Wuxi GuoSheng Bio-Engineering Co., Ltd. (Wuxi, China). The cell culture plates (96-well) and the cell culture flasks were bought from Wuxi Nest Biological Technology Co., Ltd. (Wuxi, China). The nitrocellulose (NC) membrane, CB-SB08 glass fiber membrane, polyvinylchloride (PVC) backing material, and an absorbent pad were bought from GoldBio Technology Co., Ltd. (Shanghai, China). A strip-cutting instrument (HM2025) was purchased from Shanghai Gold Bio-Pharmaceutical Technology Co., Ltd. (Shanghai, China). A strip reader, which was manufactured by Determine Bio-Tech Co., Ltd. (Wuxi, China), was also obtained.

### 2.2. Preparation of IMP Hapten, Protein Conjugates, Immunization, mAb Production, and Characterization of ELISA Method

Briefly, IMP (2 g), mercaptopropionic (1.66 g), and Cs_2_CO_3_ (25.5 g) were diluted in dimethyl adipate (30 mL) at 120 °C. After 16 h of reaction, the mixture was cooled to room temperature. Next, 50 mL of tetrahydrofuran was added, and the mixture was filtered to obtain the crude solid product (IMPA, 30 g, Figure 1b.).

The IMPA active group was synthesized using the EDC method. Briefly, a mixture of IMPA (2.5 mg), EDC (5 mg), and NHS (3 mg) in DMF (300 µL) (solution A) was stirred for 6 h at room temperature, then, solution B, which consisted of KLH (5 mg/mL, 1 mL) that was diluted with borate buffer (BB, 1 mL), was added dropwise into solution A, and reacted for 24 h at room temperature to obtain the conjugate IMPA-EDC-KLH mixture. The next step was dialysis, in which the complete antigen and unconjugated small molecule hapten were separated and characterized by an ultraviolet (UV) scanning method.

IMPA (2.2 mg), EDC (4 mg), and NHS (2.5 mg) were dissolved in DMF (300 µL) (solution A) for 4–6 h at room temperature and stirred continuously. Solution B consisted of OVA (10 mg) that was dissolved in BB (2 mL), which was then added dropwise into solution A and reacted for 24 h at room temperature to obtain the conjugate IMPA-EDC-OVA mixture. The mixture was dialyzed and characterized using the UV scanning method.

We immunize the female BALB/c mice (approximately 6–8 weeks old) using IMPA-EDC-KLH to produce mAbs and the coating antigen IMPA-EDC-OVA. First, the mice were immunized using complete Freud’s adjuvant, followed by Freud’s incomplete adjuvant, every 3 weeks. The half-maximal inhibitory concentration (IC_50_) was determined using an ELISA and the highest titer was selected. Next, splenocytes were collected and fused with Sp 2/0 murine myeloma cells to produce hybridomas. To produce the optimal mAbs, the hybridomas were injected into the mice. The resulting ascitic fluid was collected, purified, and kept at −20 °C until use [25].

### 2.3. Preparation of LFA Based on AuNP-Labelled mAbs (GLM)

HAuCl_4_ (0.4 g L^−1^, 12.5 mL) was added to pure water (487.5 mL), boiled for 15 min, then sodium citrate tribasic dihydrate solution (1 mL, 1%, *w*/*v*) was added. After 30 min of boiling and continuous stirring, the heat was turned off and the mixture color became red. Afterwards, it was cooled to room temperature, and the mixture was kept at 4 °C. Finally, the AuNPs were characterized using TEM (Figure 1c).

The anti-IMP mAbs (IC_50_ = 0.27 ng mL^−1^) were conjugated to the AuNPs. In brief, we added K_2_CO_3_ (0.1 M, 40 µL) into 10 mL of AuNP solution, then the mAb solution (0.4 mL, concentration of mAb = 0.2 mg mL^−1^) was added (react 45 min). Afterwards, we added 10% BSA (*w*/*v*) as a blocking buffer. After 2 h of blocking, we centrifuged the mixture for 45 min (18,000× *g*). We discarded the resuspend and reconstituted the lower layer using resuspension buffer (tris-HCl (0.02 M), added Tween-20 (0.1%), polyethylene glycol (PEG)-20,000 (0.1%), polyvinylpyrrolidone (PVP) (1%), sucrose (5%), trehalose (4%), sorbitol (2%), mannitol (1%), NaN_3_ (0.04%), and last BSA (0.2%), 1 mL), and kept the mixture at 4 °C.

### 2.4. Assay Procedure

First, the NC membranes were laminated to the center of a plastic backing plate. Then, the NC membrane was sprayed with goat anti-mouse IgG antibody (C line) and IMPA-EDC-OVA was sprayed on the test (T) line. Afterward, we dried the strip test overnight at 37 °C.

Actual samples were used to determine the sensitivity of the LFA. The conjugated pad consisted of GLM diluted in resuspension buffer with 5% BSA by four folds. After the conjugated pad was dried at 37 °C for 24 h, all the pads, including the conjugated pad, sample pad, and absorbent pad were attached onto the plate and we cut the strips (3 mm). Then, we dropped the sample solution (100 µL) into the sample pad. After 7 min of reaction, the color developed, and we read the color intensity using the strip reader.

To determine the sensitivity of LFA, the IMP standards were diluted in PBS (0.01 M) with final concentrations of 0.1, 0.25, 0.5, 1, 2.5, 5, and 10 ng mL^−1^ for GLM. We compared the color intensity on the T line of the negative and positive results with the naked eye to develop the cut-off value of the GLM strip.

### 2.5. Sample Analysis

We used a simple pre-treatment to detect IMP in Chinese cabbage, cowpea, pear, and apple samples. Briefly, we weighed 1 g of the samples and placed them into 10 mL centrifuge tubes, then extraction buffer (5% methanol added 1% Rhodasurf On-870, 3 mL) was added and homogenized (5 min). A total of 100 µL of the mixture sample was added into the dilution buffer (0.01 M PBS added 1% Rhodasurf On-870, 200 µL).

### 2.6. Accuracy and Precision of the Developed LFA for Spiked Samples

Accuracy and precision were assessed by detecting Chinese cabbage, cowpea, pear, and apple samples spiked with three IMP concentrations. The precision of the method was developed by calculating the coefficient of variation (CV = standard deviation (SD)/mean). The T/C values (color intensity of the T line to that of the C line) of the spiked samples were calculated using a strip reader, and to calculate recovery we used the following formula: (concentration measured/concentration spiked) × 100. Six replicates at each concentration were assayed.

## 3. Results and Discussion

IMP was successfully double extracted from soil and quantified by reversed-phase HPLC. The first extraction based on the Soxhlet method, which was a mixture of acetone and hexane, was performed, and was continued with the second extraction (acetonitrile, methanol, and water). The relative SD of the liquid extraction and the Soxhlet extraction methods was 1.9–5.6% and the LOD results were 0.08 and 0.06 mg kg^−1^, respectively [31]. The determination of IMP in white pine using a GC/MS method with a limit of quantitation (LOQ) of 0.01 ppm (10 pg injected) was well performed [32]. The sample preparation was based on the QuEChERS (quick, easy, cheap, effective, rugged, and safe) method using UHPLC–MS/MS. Following the optimization and the validation, the LOQ values were 0.25, 0.5, and 1 µg kg^−1^ for honey, bees, and beeswax, respectively [33].

LC–MS/MS is currently the preferred method for the detection of insecticide residues. One study [15] determined a QuEChERS assay using LC–MS/MS to detect the residues of five neonicotinoid pesticides (acetamiprid, clothianidin, IMP, thiacloprid, and thiamethoxam) in propolis (a resin-like material that is made by bees from the buds of poplar and cone-bearing tress) with a LOD of 0.2–4.4 ng g^−1^ and a LOQ of 0.8–14.7 ng g^−1^. A combined method of multi-plug filtration cleanup (m-PFC) and LC–MS/MS was established to simultaneously detect 14 pesticides, including IMP, with a LOD of 3–50 µg Kg^−1^ [34]. Even though chromatography is highly sensitive, there are inherent drawbacks, such as time-consuming procedures, the complex pre-treatment of the samples, expensive instruments, and the need for specialized personnel. The aim of this study was to develop LFA as an on-site, sensitive, and fast assay to detect IMP residues, which has been tested with the use of Chinese cabbage, cowpea, pear, and apple samples.

### 3.1. Characterization of IMPA-EDC-KLH and IMPA-EDC-OVA

The UV–visible absorption spectra of IMPA-EDC-KLH and IMPA-EDC-OVA are shown in Figure 1d,e. The maximum peaks of KLH and IMPA were 278 and 269 nm, respectively, while the IMPA-EDC-KLH had a strong absorbance peak of 276 nm, which was shifted, and the peak type was superimposed. In addition, the absorbance peak of IMPA-EDC-OVA obviously shifted. The obvious peak shifts between the IMPA-EDC-KLH and IMPA-EDC-OVA and KLH, OVA, and IMPA indicate the successful synthesis of antigens.

### 3.2. Assessment of Results and Characterization of the LFIA

Using an ELISA method, the standard curves in this study are shown in Figure 1f, and the specificity of the mAb with the use of an ELISA is listed in Table 1. The LFA apparatus consists of four parts (Figure 2a).

The interaction between the analyte (sample) and the conjugate (labelled antibody) will migrate towards the NC membrane. Then, slowly, a red-colored band on the C and T lines will develop, depending on the concentration of IMP. A negative result (no IMP standard) is indicated by two red-colored bands at both of the lines. Only a red-colored band on the C lines indicates a positive result. The lack of a red-colored band at both lines indicates that the test procedure is invalid (Figure 2b). The strip reader was used to determine the color intensity that developed on both of the lines (Figure 2c).

### 3.3. Performance of the LFA

In order to function as the reaction matrix in the LFA system, the materials must have hydrophobic and consistent flow characteristics. A rewetting agent was added in order to render the material hydrophilic. Not every protein might be suitable with every surfactant. We optimized the process by adding an extra amount of PVP, PEG, BSA, and Rhodasurf^®^ ON-870 by 5% into the resuspension buffer. The results of the LFA were obtained in about seven minutes (Figure 3a,b). The resuspension of the buffer containing 5% BSA was carried out in a further experiment due to the stability of its color intensity on both of the lines. To verify the strip, IMP was added into the PBS (0.01 M) in final concentrations of 0, 0.1, 0.2, 0.5, 1, 2, 5, and 10 ng mL^−1^ (Figure 3b). The color intensity was weakened when we added IMP with a higher concentration and with no color at 10 ng mL^−1^.

### 3.4. Detection of IMP Spiked in Chinese Cabbage, Cowpea, Pear, and Apple samples

We bought the Chinese cabbage, cowpea, pear, and apple samples at Wuxi market (China). In order to verify the absence of IMP, all of the samples were analyzed using LC–MS (Appendix A). All four of the samples were spiked with IMP at 0, 0.2, 0.5, 1, 2, 5, 10, and 20 ng mL^−1^ (Figure 3c). For each negative sample, deep-red-colored bands developed on the C and T lines and cut off at 20 ng mL^−1^. A calibration curve was constructed by plotting the T/T_0_ ratio (color intensity of T = added IMP samples (T line) and T_0_ = negative samples (C line)) against the standard concentrations of IMP for each concentration (Figure 3d).

In order to confirm the sensitivity and the accuracy of the LFA, the samples (Chinese cabbage, cowpea, apple, and pear) were spiked with different concentrations of IMP (Table 2). The range of recovery and the CV of the LFA for the determination of IMP residues on the Chinese cabbage, cowpea, apple, and pear samples were 99.3–102.2%/4.1–6.4%, 96–104.7%/4.3–9.3%, 96–103%/3.3–8.9%, and 97.2–101.7%/3.6%–5.6%, respectively. These results indicate that the GLM-based LFA was a quick and sensitive assay to detect IMP residues on fruits and vegetables.

## 4. Conclusions

We have developed a highly sensitive mAb for the detection of IMP. The proposed LFA is a rapid and sensitive assay to detect IMP residues on fruits and vegetables with cut-off values of 20 ng mL^−1^. Based on the recovery and the CV values of the spiked samples, the proposed LFA based on the GLM strips is a fast, sensitive, reliable, inexpensive, and relatively easy method for the on-site determination of IMP residues on fruits and vegetables.

## Figures and Tables

**Figure 1 foods-12-00512-f001:**
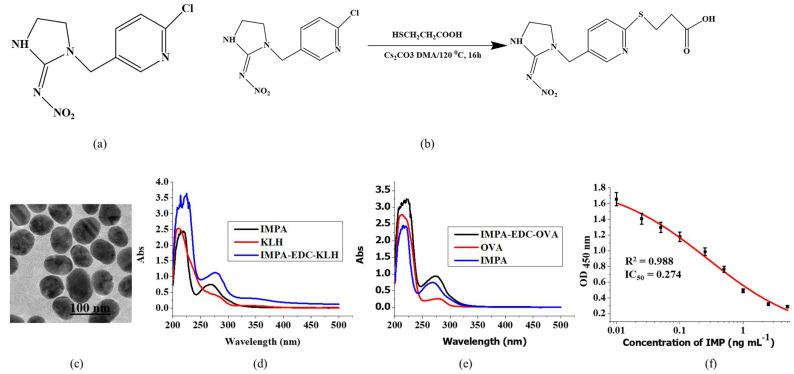
(**a**) Structure of IMP. (**b**) Preparation of IMP hapten. (**c**) Result of AuNPs by transmission electron microscopy (TEM). A diagram of the use of UV–vis spectra of IMP immunogens (**d**) and IMPA-EDC-OVA (**e**). Standard curves of IMP detection using ELISA (**f**).

**Figure 2 foods-12-00512-f002:**
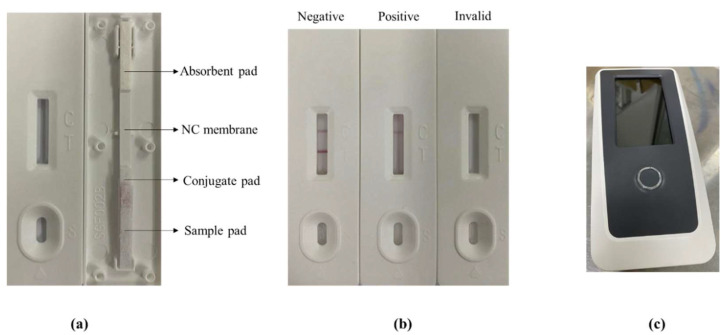
Parts of the LFA strip (**a**). Diagram of test results (**b**). Strip reader (**c**).

**Figure 3 foods-12-00512-f003:**
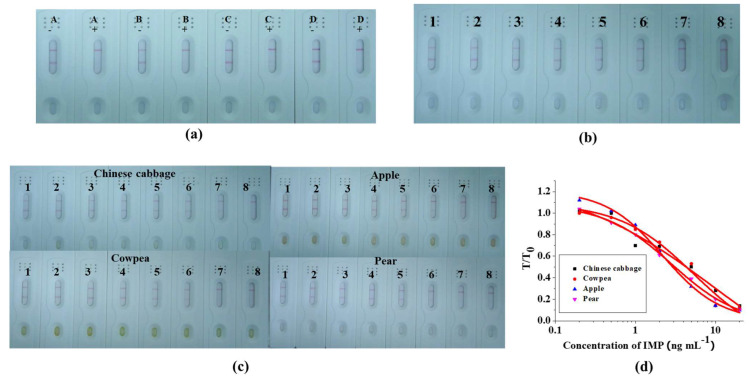
Results of the optimization of the strip test with extra added A = PVP; B = PEG-2000; C = BSA; and D = Rhodasurf^®^ On-870 in the resuspension buffer (**a**). negative (−) and positive (+) = 5 ng mL^−1^. The strip used to detect IMP in 0.01 M PBS (**b**), 1–8 represents concentrations of 0, 0.1, 0.2, 0.5, 5, 2, 5, and 10 ng mL^−1^. Result after optimization to determine IMP in Chinese cabbage, cowpea, apple, and pear samples; (1–8) represents concentrations of 0, 0.2, 0.5, 1, 2, 5, 10, and 20 ng mL^−1^ (**c**). Quantitative curve of binding rate (T/T_0_) against concentration of IMP (**d**). T = sample containing IMP; T_0_ sample without IMP.

**Table 1 foods-12-00512-t001:** Results of cross-reactivity using imidacloprid mAb.

Chemical Compound	IC_50_ (ng mL^−1^)	Cross-Reaction (%)
Imidacloprid	0.274	100
Clothianidin	40.016	0.685
Thiacloprid	50.121	0.547
Nitenpyram	>1000	<0.027
Thiamethoxam	>1000	<0.027
Acetamiprid	>1000	<0.027
Nitenpyram	>1000	<0.027

**Table 2 foods-12-00512-t002:** Results of IMP detection in Chinese cabbage, cowpea, apple, and pear samples using LFA (*n* = 6).

Samples	Spiked IMP (ng mL^−1^)	GLM Strip		
		Recovery ± SD ^a^ (%)	CV ^b^ (%)	Qualitative Results
Chinese cabbage	5	99.3 ± 0.3	5.1	−
	10	102.2 ± 0.6	6.4	±
	20	99.4 ± 0.8	4.1	+
Cowpea	5	96 ± 0.3	5.2	−
	10	104.7 ± 0.9	9.3	±
	20	98.9 ± 0.9	4.3	+
Apple	5	96 ± 0.4	8.9	−
	10	100.3 ± 0.9	8.5	±
	20	100.3 ± 0.7	3.3	+
Pear	5	101.7 ± 0.3	5.6	−
	10	97.2 ± 0.6	5.6	±
	20	101.5 ± 0.7	3.6	+

^a^ SD, standard deviation. ^b^ CV, coefficients of variation. −, negative: the concentration of IMP was below 5 ng mL^−1^; ±, weakly positive: the concentration of IMP was in the range of 5–15 ng mL^−1^; +, positive: the concentration of IMP exceeds 20 ng mL^−1^.

## Data Availability

Not applicable.

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
