# Peer review of "A Rapid Immunochromatographic Method Based on Gold Nanoparticles for the Determination of Imidacloprid on Fruits and Vegetables"

_foods, 2023, doi:10.3390/foods12030512_

Round 1

Reviewer 1 Report

When reviewing the work, I can determine that it is of interest beyond the food areas, including agribusiness, therefore, I consider that it can be improved a lot.

The points are as follows:

1. The reference format delivered does not correspond to what was requested by the journal, so leave all highlighted, in yellow, the format described for authors is numeral. Also the bibliography there are different formats should be reviewed.

2. There is a great use of the acronym in the titles (IMP) I suggest the use of the word Imidacloprid in the titles and Table 1. 

3. In the conclusions, the economics of the technique are mentioned, but the general costs of application and manufacturing of the "product" are not clear to me.

4. The fruits that were used in this study, the species or variety used is not specified.

5. in point 173 the unit ng/g must be presented as ng × g1 

6. Point 335 Procambarus clarkii should be in italics.

Reviewer 2 Report

The manuscript of Steven Suryoprabowo et al. describes the development of a gold nanoparticles-based lateral flow immunoassay for the detection of imidacloprid in real matrices.

The manuscript would be of interest for the Foods readership. However, in its current form it needs revisions.

Below some suggestion to improve the manuscript.

First of all, I would like to invite the authors to deeply revise the English, especially grammar, verbal tense, etc., since the manuscript is quite difficult to read.

Just some non-exhaustive examples:

Abstract. Line 9 “used”. Line 10, the wrong use of “however”. Line 11 “on-site monitoring” of what? Line 13, the use of “is” and “was”.

Introduction. Line 24, the wrong use of “however”. Line 27, “a fast and have a…”. Lines 40,41 “Although these methods are perfectly sensitive and steady, and all are relatively complex, needs time to operate, labor intensive, and expensive” (please revise the whole sentence). Line 53 “An ELISA is sensitive method”. Line 58 “ELISA method need”. Line 61 “lateral flow assays (LFAs) is a rapid…”. Lines 70,71 “We proposed that LFA is the first rapid assay to detect IMP residues on fruits and vegetables”

Materials and Methods. Lines 146 “…and apple, we using…”

Results and Discussion. Lines 158,159 “IMP was successfully double extracted from soil and quantified was by reversed-158 phase HPLC.” Line 176 “The proposed to develop LFA”. Line 188 “was shown”. Line 196 “Upon interaction, the interaction”. Line 209 “because the stable”. Line 221 “we were bought”.

Conclusions. Line 241 “We developed mAb”

Considerations not related the English improvement.

Please improve the quality of image 1.

Page 2 lines 57,58. Please specify when the LOD is 0.1 ng/mL and when it is 0.5 ng/mL.

Page 2 lines 64,65. “Some LFAs use antibodies labelled with gold nanoparticles (AuNPs), while others employ fluorescent-microspheres….”. Since most of the LFAs used AuNPs as label (not just some), as quantitatively reported in https://doi.org/10.3390/s21155185, I suggest to use “Most of the LFAs uses antibodies labelled with gold nanoparticles (AuNPs) [https://doi.org/10.3390/s21155185,], while others employ fluorescent-microspheres….”.

Page 3 line 86. Please use “absorbent pad” instead of “absorbance pad”.

Please add the ethical statement regarding the animal use.

Page 4 lines 152,153. “The accuracy and precision of the method was developed by calculating the coefficient of variation (CV= standard deviation (SD) / mean).” Please revise, since the CV is used mainly to evaluate the precision, while the recovery is used to evaluate the accuracy.

Page 5 lines 190-192 “The NC membrane was attached to a PVC bottom plate. The other pads (the sample pad, conjugate pad and absorbent pad) were also attached on both ends to overlap the NC membrane.” I suggest to remove this part from the Results and Discussion section.

Page 6 line 204. To this reviewer it is not clear what the author mean with “The performance of the LFA was determined using a hydrophobic material.”

Page 6 lines 225,226 “A calibration curve was constructed by plotting the standard concentrations of IMP against the T/T0 ratio”. Please invert the order in the sentence, since the first one (the one that is plotted against something else) is usually related to the y-axis, while the second one to the x-axis.

Reviewer 3 Report

The paper describe a rapid method by immunochromatography for the determination of Imidacloprid on fruits and vegetable, the paper need some revision in the following points:

 1. Introduction - the author should explain better the advantages and disadvantages in the use of this method respect to the other methods cited in the lines 31- 42. It is not very clear the advantages because the preparation of the reagent spent long time.

Lines 158-175 should be move on in the introduction.

The limit of quantification, matrix effect and a comparison of the analysis results with other techniques as LC/MS/SMS should be describe in the paper.

The paper describe a rapid method by immunochromatography for the determination of Imidacloprid on fruits and vegetable, the paper need some revision in the following points:

1. Introduction - the author should explain better the advantages and disadvantages in the use of this method respect to the other methods cited in the lines 31- 42. It is not very clear the advantages because the preparation of the reagent spent long time.

Lines 158-175 should be move on in the introduction.

The limit of quantification, matrix effect and a comparison of the analysis results with other techniques as LC/MS/SMS should be describe in the paper.

Round 2

Reviewer 2 Report

The authors have addressed the issues raised by the reviewers improving their manuscript.